# Materializing efficient methanol oxidation via electron delocalization in nickel hydroxide nanoribbon

Xiaopeng Wang[1], Shibo Xi[2,11], Wee Siang Vincent Lee [1,11], Pengru Huang[1,3], Peng Cui[4], Lei Zhao [5], Weichang Hao [6], Xinsheng Zhao[4], Zhenbo Wang [5], Haijun Wu [1], Hao Wang [7], Caozheng Diao[8], Armando Borgna[2], Yonghua Du [9✉], Zhi Gen Yu [10✉], Stephen Pennycook[1✉] & Junmin Xue [1✉]

Achieving a functional and durable non-platinum group metal-based methanol oxidation catalyst is critical for a cost-effective direct methanol fuel cell. While $Ni(OH)_2$ has been widely studied as methanol oxidation catalyst, the initial process of oxidizing $Ni(OH)_2$ to NiOOH requires a high potential of 1.35 V vs. RHE. Such potential would be impractical since the theoretical potential of the cathodic oxygen reduction reaction is at 1.23 V. Here we show that a four-coordinated nickel atom is able to form charge-transfer orbitals through delocalization of electrons near the Fermi energy level. As such, our previously reported periodically arranged four-six-coordinated nickel hydroxide nanoribbon structure (NR-$Ni(OH)_2$) is able to show remarkable methanol oxidation activity with an onset potential of 0.55 V vs. RHE and suggests the operability in direct methanol fuel cell configuration. Thus, this strategy offers a gateway towards the development of high performance and durable non-platinum direct methanol fuel cell.

[1] Department of Materials Science and Engineering, National University of Singapore, Singapore 117575, Singapore. [2] Institute of Chemical and Engineering Sciences, Agency for Science, Technology and Research, Singapore 627833, Singapore. [3] Guangxi Collaborative Innovation Center of Structure and Property for New Energy, Guangxi Key Laboratory of Information Materials, School of Material Science and Engineering, Guilin University of Electronic Technology, Guilin 541000, China. [4] School of Physics and Electronic Engineering, Jiangsu Normal University, Jiangsu Sheng 221100, China. [5] School of Chemistry and Chemical Engineering, Harbin Institute of Technology, Heilongjiang Sheng 150006, China. [6] School of Physics, Beihang University, Beijing 100191, China. [7] Department of Mechanical Engineering, National University of Singapore, Singapore 117575, Singapore. [8] Singapore Synchrotron Light Sources (SSLS), National University of Singapore, Singapore 117603, Singapore. [9] National Synchrotron Light Source II, Brookhaven National Laboratory, Upton, NY 11973, USA. [10] Institute of High Performance Computing, Agency for Science, Technology and Research, Singapore 138632, Singapore. [11] These authors contributed equally: Shibo Xi, Wee Siang Vincent Lee. ✉email: ydu@bnl.gov; yuzg@ihpc.a-star.edu.sg; steve.pennycook@nus.edu.sg; msexuejm@nus.edu.sg

Methanol is an extremely attractive fuel due to its high energy density and its ease of handling[1–6]. As a result, there is a concerted effort in the advancement in $CO_2$ reduction efficiency to produce more methanol for energy storage, and to reduce the increasing $CO_2$ pollutant generated in society[7]. With the increasing methanol availability, the challenge will be to develop an efficient device that can effectively convert methanol into electrical power. As such, direct-methanol fuel cells (DMFCs), being one such device that utilizes methanol as fuel, has attracted significant research interest due to its high portability and its promise of high energy density[1–6,8–14]. While DMFCs are poised as one of the future energy sources in a hypothesized carbon-neutral economy, there is still a huge deficit in the development of a suitable electrocatalyst which demands considerable research attention. The main underlying factor for this material selection deficit is in the anodic reaction of DMFCs where the methanol oxidation reaction (MOR) is kinetically sluggish due to its six-electron transfer process[8–13]. Thus, despite the significant effort devoted in the development of MOR catalysts in recent decades, platinum group metals (PGMs) remain the only known MOR catalysts that are reported to trigger methanol oxidation during DMFC operation[8]. This is largely due to the ability of PGMs to operate at *ca.* 0.4–0.6 V vs. RHE, which is below the theoretical oxygen reduction reaction (ORR) potential of 1.23 V vs. RHE. While PGMs have been used in DMFCs, their practicality as functional MOR catalyst is stifled by their high cost[5,8,11]. Hence, it is of significant importance to reduce the reliance in PGMs by developing an alternative material as a durable and cost-effective catalys

Ni(OH)$_2$ has been widely studied as an electrocatalyst for various oxidation reactions which include MOR[12,13]. Before the MOR process, Ni(OH)$_2$ has to be first electrochemically oxidized to NiOOH at a potential above 1.35 V vs. RHE in order to exhibit MOR activity[12]. Such potential would be impractical as a DMFC anode since it is higher than its cathodic ORR potential (1.23 V vs. RHE). Thus, the need to oxidize Ni(OH)$_2$ to NiOOH at high potential in order to trigger limited MOR activity in a three-electrode electrochemical environment suggests that such material cannot be considered as functional MOR catalysts in DFMC configuration. This inadequacy of traditional Ni(OH)$_2$ as a functional MOR catalyst is primarily due to the forbidden electron transfer on valence band near Fermi energy level in a traditional six-coordinated Ni atom, which ultimately leads to the ineffectiveness of Ni(OH)$_2$ in oxidizing methanol. When $Ni^{2+}$ is oxidized to $Ni^{3+}$, an available charge-transfer orbital is formed which is then available for the electron transfer process during methanol oxidization to $CO_2$. However, it is fundamentally challenging to create available charge-transfer orbitals near the Fermi energy level in traditional Ni(OH)$_2$ due to these six-coordinated Ni atom without subjecting the material to impractical potential. Interestingly, it is unveiled in this work that when the coordination number of Ni is reduced to four, electrons can be delocalized near the Fermi energy level which ultimately generates an available charge-transfer orbital to facilitate the six-electron transfer process during MOR. Despite the importance of four-coordinated Ni atoms in the establishment of charge-transfer orbital due to delocalization, it is shown that this four-coordinated Ni atom must be paired to a neighboring six-coordinated Ni atom in order to achieve efficient methanol oxidation. As such, based on this insight, it is suggested that Ni(OH)$_2$ with four/six coordinated Ni atoms would be highly suitable as a functional MOR catalyst.

Here we show that our previously reported strain-stabilized non-stochiometric Ni(OH)$_2$ nanoribbon structure with alternating 4/6-coordinated Ni edge atoms (NR-Ni(OH)$_2$)[15] is a potential functional MOR catalyst. It is revealed that NR-Ni(OH)$_2$ is able to exhibit MOR activity in alkaline electrolyte at a significantly lower onset potential of 0.55 V vs. RHE, which is close to that of platinum anode. It is proven by both theoretical and experimental results that this highly efficient MOR activity is due to the alternating four-and six-coordinated Ni atoms. The operability of NR-Ni(OH)$_2$ is firstly assessed by determining the open circuit voltage (OCV) of the assembled DMFC. Unlike the DMFCs assembled with traditional Ni(OH)$_2$ (negligible OCV), the NR-Ni(OH)$_2$ assembled DMFC is able to achieve a serviceable OCV of 0.58 V, which is reasonably close to that of its counterparts assembled with traditional Pt (OCV of 0.65 V) and PtRu (OCV of 0.70 V). Our impressive result is, to the best of our knowledge, the first report of non-PGM MOR catalyst that can achieve an operable OCV in DFMC configuration. Furthermore, the initial assessment of NR-Ni(OH)$_2$ in DFMC configuration has shown that the device was able to achieve durable operation with zero CO poisoning. Thus, based on these positive indications of DFMC operability, it is expected that with more research, NR-Ni(OH)$_2$ can be further optimized to become an important non-PGM MOR catalyst in DFMC.

## Results

**MOR activity of NR-Ni(OH)$_2$.** To properly trigger the operation of a DMFC, the anode potential is required to be smaller than the cathode potential (Supplementary Fig. 1a). Thus, the MOR catalyst must show the capacity to oxidize methanol into $CO_2$ below theoretical ORR potential i.e., 1.23 V vs. a reversible hydrogen electrode (RHE). Here, the MOR activity of NR-Ni(OH)$_2$ is evaluated by conducting CV scans from 0 to 1.2 V in 1 M KOH + 1 M CH$_3$OH electrolyte solution using a three-electrode system. Traditional Ni-based catalysts, including α/β-Ni(OH)$_2$, Ni metal, NiFe, are used as controls (Supplementary Figs. 1b–e and 2). All potentials recorded are vs. RHE. To avoid the effect of Pt on MOR measurement, graphite paper was used as counter electrode. The sample after MOR measurement is further analyzed using inductively coupled plasma (ICP) chemical analysis whereby a low Pt content of 0.001% in the NR-Ni(OH)$_2$. Thus, based on this result, Pt effect on the MOR activity could be ignored. The MOR onset potential of the sample is as low as 0.55 V, with two obvious oxidation peaks at 0.89 V (forward) and 0.79 V (backward), respectively (see Fig. 1a, in solid), while no oxidation peaks were seen in traditional Ni-based catalysts in potential range from 0 to 1.23 V (Supplementary Fig. 1b–e and 2). It is noted that the MOR oxidation peak 0.89 V is well below the theoretical ORR potential of 1.23 V, indicating that NR-Ni(OH)$_2$ has promising application for a DMFC full cell. In contrast, no oxidation peaks are detected when the CV scan is conducted in 1 M KOH (dashed line, Fig. 1a). Thus, it can be safely concluded that the observed oxidation peaks arise from methanol oxidation. Furthermore, the sample shows excellent cycle stability, without obvious shift in oxidation peak position or decrease in current density after 3000 CV cycles (Fig. 1b). As a comparison, the MOR activity of Pt/C was also recorded under the same test condition (Fig. 1c), showing an onset potential of 0.54 V and two methanol oxidation peaks at 0.79 V (forward) and 0.74 V (backward), respectively, but with much lower peak current densities compared to that of NR-Ni(OH)$_2$. Moreover, the stability indicating the retained activity of electrocatalysts after long-term operation, was measured through chronoamperometric measurement using the previous method[13]. Over 3600 s operation NR-Ni(OH)$_2$ exhibits 16% in activity, while Pt/C only has 4% activity, suggesting that NR-Ni(OH)$_2$ has better stability than Pt/C (Supplementary Fig. 3).

Next, open-circuit voltage (OCV) and stability measurements were used to identify the feasibility of NR-Ni(OH)$_2$ in DMFCs. The OCV measurement was conducted using the previous method[14], being benchmarked with α/β-Ni(OH)$_2$, Ni metal, NiFe

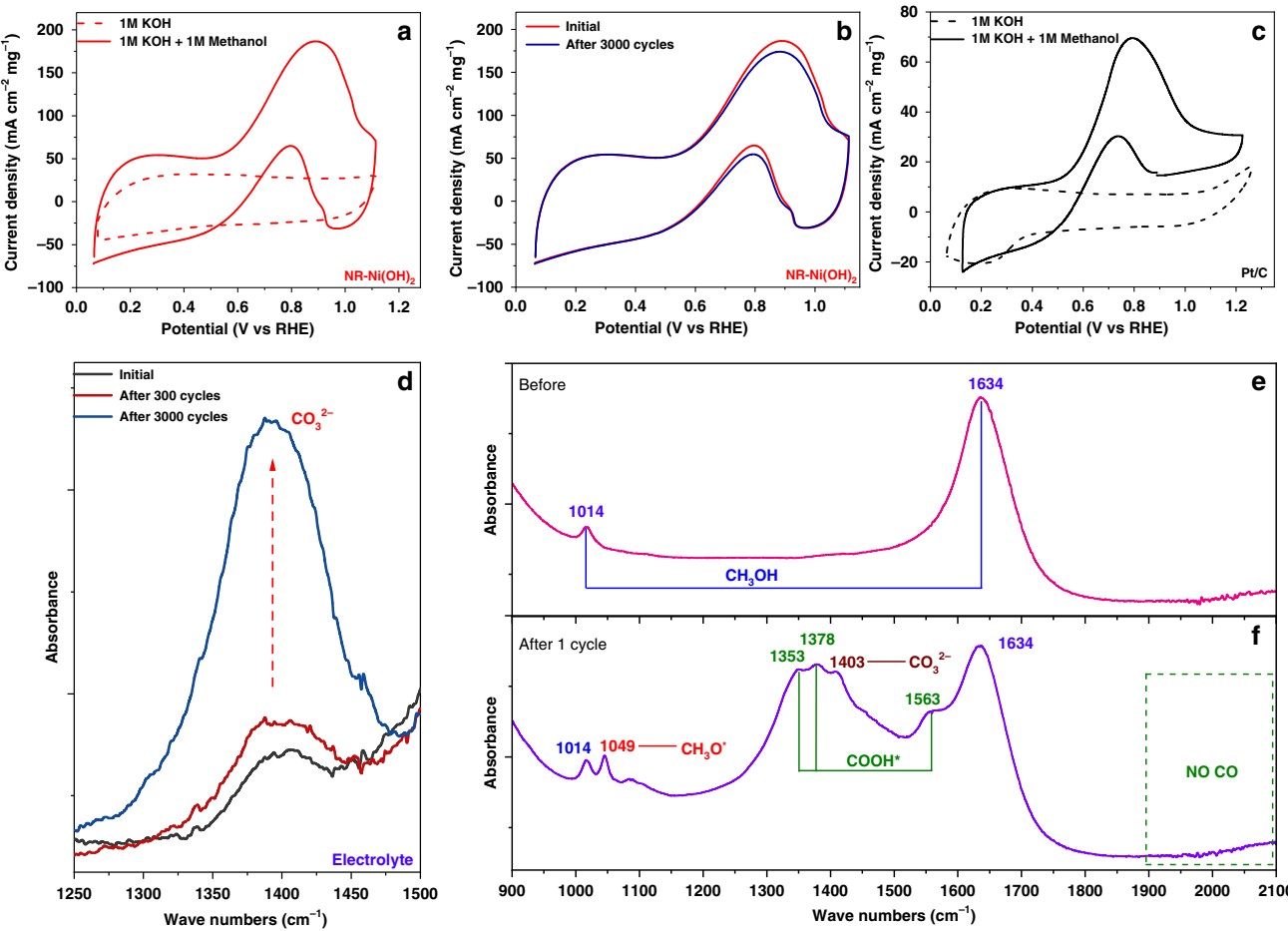

**Fig. 1 Catalytic activity of NR-Ni(OH)$_2$ in 1 M KOH + 1 M CH$_3$OH. a** Cyclic voltammograms of NR-Ni(OH)$_2$ recorded in 1 M KOH (dashed) and 1 M KOH + 1 M CH$_3$OH (solid) solutions at a scan rate of 20 mV s$^{-1}$; **b** cycle stability of NR-Ni(OH)$_2$ recorded in 1 M KOH + 1 M CH$_3$OH solution after 3000 cycles; **c** cyclic voltammograms of Pt/C recorded in 1 M KOH (dashed) and 1 M KOH + 1 M CH$_3$OH (solid) solutions at a scan rate of 20 mV s$^{-1}$; **d** FTIR spectra of the electrolytes upon different CV cycles; **e** FTIR spectrum of electrode surface before CV scan; **f** FTIR spectrum of electrode surface after one CV cycle.

LDH, standard Pt/C, and PtRu/C catalyst. As shown in Supplementary Table 1, nearly no electric output was seen for α/β-Ni(OH)$_2$, Ni metal and NiFe LDH catalysts, while high OCV values (OCV > 0.55 V) were found in NR-Ni(OH)$_2$, Pt/C and PtRu/C catalysts. These results indicated, NR-Ni(OH)$_2$ could trigger the operation of DMFCs. A DMFC full cell prototype was further investigated with NR-Ni(OH)$_2$ and Pt/C as the MOR and ORR catalysts, respectively, as shown in Supplementary Fig. 4. At a constant discharge current of 1.0 mA cm$^{-2}$, the cell delivers a more stable output voltage, while the DMFC using Pt/C as both anode and cathode shows an obvious decrease from 0.48 V to 0.33 V, implying that NR-Ni(OH)$_2$ has better stability than Pt/C, which agrees with the electrochemical durability results (Supplementary Fig. 3). The slight decrease in output voltage of DMFC with the use of NR-Ni(OH)$_2$ may be due to the exchange of OH$^-$ with carbonate (CO$_3^{2-}$) and biocarbonate (HCO$_3^-$), which can greatly decrease the conductivity of membrane[16]. All these results confirm that NR-Ni(OH)$_2$ is workable in a DMFC full cell.

Fourier transform infrared spectra (FTIR) is employed to identify the methanol oxidation process on NR-Ni(OH)$_2$. In the electrolyte solution, the concentration of CO$_3^{2-}$ continuously increased with increasing cycle number (Fig. 1d), indicating that methanol could be fully oxidized into CO$_2$[17]. To identify the possible residual intermediates, the electrode surface after one CV cycle is characterized using FTIR, while the electrode surface without any CV scan is used as a control (Fig. 1e, f). Residual

intermediates of CH$_3$O* and COOH* were detected upon one CV scan (Detailed FTIR analysis is provided in Supplementary Fig. 5)[17–20]. The presence of CH$_3$O* and COOH* as residual intermediates suggests that the reaction mechanism of MOR catalyzed by NR-Ni(OH)$_2$ is different from that of Pt/C, in which CO* and COOH* are usually the residual intermediates[20].

**MOR mechanism of NR-Ni(OH)$_2$.** In-situ FTIR was conducted to track the changes in the respective amounts of CH$_3$O* and COOH* upon one CV cycle with an applied voltage from 0 to 1.2 V, which are represented by the changes in band absorbance intensity (Fig. 2a, b). From 0 to 0.6 V, the amount of CH$_3$O* is slightly decreased, indicating only little oxidation occurred but the major reaction has not yet started. Above 0.6 V, the amount of CH$_3$O* dramatically increased, suggesting its fast accumulation and thus the start of the reaction, which agrees well with the onset potential of the forward oxidation peak shown in Fig. 1a. COOH is decreased in amount from 0 to 1.2 V, but with a sudden increase in consumption rate above 0.6 V, further confirming that the reaction starts above 0.6 V. To gain an in-depth understanding of the methanol oxidation process on the nanoribbons, density functional theory was used to calculate the energy landscapes controlling the decomposition of intermediates. Based on the optimized layered unit cell of Ni(OH)$_2$ and the experimental results, we constructed a Ni(OH)$_2$ nanoribbon supercell (Supplementary Fig. 6). Ni atoms

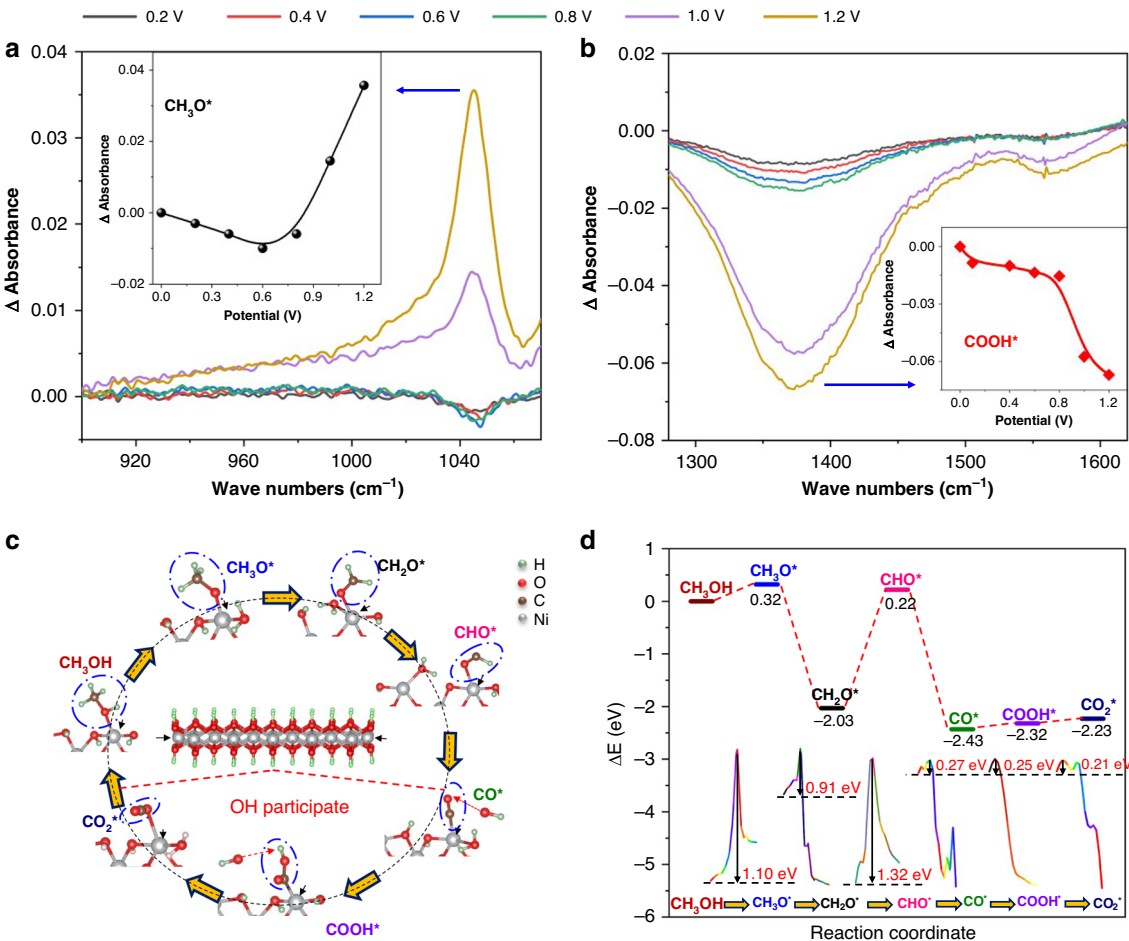

**Fig. 2 MOR mechanism of NR-Ni(OH)$_2$. a** In-situ FTIR spectra of NR-Ni(OH)$_2$/solution in the wave number ranges from 900 to 1070 cm$^{-1}$ (inset representing the detailed variations of CH$_3$O$^*$ with potential); **b** In-situ FTIR spectra of NR-Ni(OH)$_2$/solution in the wave number ranges from 1280 to 1620 cm$^{-1}$(inset showing the detailed variations of COOH$^*$ with potential). **c** Schematic of the MOR mechanism for NR-Ni(OH)$_2$; **d** Calculated reaction free energy and energy barriers during MOR.

within the ribbon are fully saturated by six OH, while the four-coordinated Ni atoms at the edges are bonded to four OH, leaving two dangling bonds. These unsaturated Ni atoms are thus able to provide the active sites for methanol adsorption. Furthermore, based on the simulation results, the methanol adsorption energy at the four-coordinated Ni atoms at the edges is much lower than at saturated Ni atoms within the ribbon (detailed analysis in Supplementary Fig. 7), which further suggests its thermodynamic feasibility. In the subsequent dehydrogenation steps, O–H and C–H chemical bonds are broken with the participation of an adjacent six-coordinated Ni atom, and the proton absorbs on the neighboring unsaturated OH (two-coordinated OH, shared by four and six- coordinated Ni atoms, as the yellow arrow indicates in the supporting information) (Fig. 2c). In other words, the four-coordinated Ni atom is responsible for efficient methanol adsorption, and the subsequent conversion of methanol to carbon dioxide is very efficiently achieved as those four-coordinated Ni atoms are located next to a six-coordinated Ni atom. Based on our calculated energy landscape, a MOR pathway of CH$_3$OH$^*$ → CH$_3$O$^*$ → CH$_2$O$^*$ → CHO$^*$ → CO$^*$ → COOH$^*$ → CO$_2$$^*$ is proposed (Fig. 2c, d). Details of the MOR activity calculations are provided in the Supplementary Discussions. Due to our periodically arranged four-six-coordinated Ni atoms, methanol absorption and its subsequent conversion to CO$_2$ is highly efficient. This also explains why β-Ni(OH)$_2$ and α-Ni(OH)$_2$ show negligible

MOR activity, while NR-Ni(OH)$_2$ exhibits remarkable MOR activity in the alkaline electrolyte (Supplementary Fig. 2).

It is worth noting that the dehydrogenation steps where the O–H/C–H chemical bonds are broken, require relatively higher energy barriers than those needed in the subsequent steps (Fig. 2c). Thus, these steps are considered as the rate-determining steps, agreeing well with the in-situ FTIR results. It is also noted that there is a potential difference between the forward (0.89 V) and backward (0.79 V) oxidation peaks, which is due to the change in the rate-determining step from O–H to C–H bond breaking during the dehydrogenation process (Supplementary Fig. 8). More details of the change in the rate-determining step are given in Supplementary Discussions. Note that the CO$^*$ oxidation step (CO$^*$ → COOH$^*$) is immediate as it has a much lower energy barrier as compared to CH$_3$OH dehydrogenations. This is in good agreement with the in-situ FTIR result, as there is no detectable CO attached on the electrode surface after applying a voltage from 0 to 1.2 V (Supplementary Fig. 9). Thus, based on these collective results, there is negligible CO poisoning for NR-Ni(OH)$_2$.

**Charge-transfer orbitals formation through delocalization of electrons near the Fermi energy level.** Previous research indicated that charge transfer in the electrocatalytic reaction for

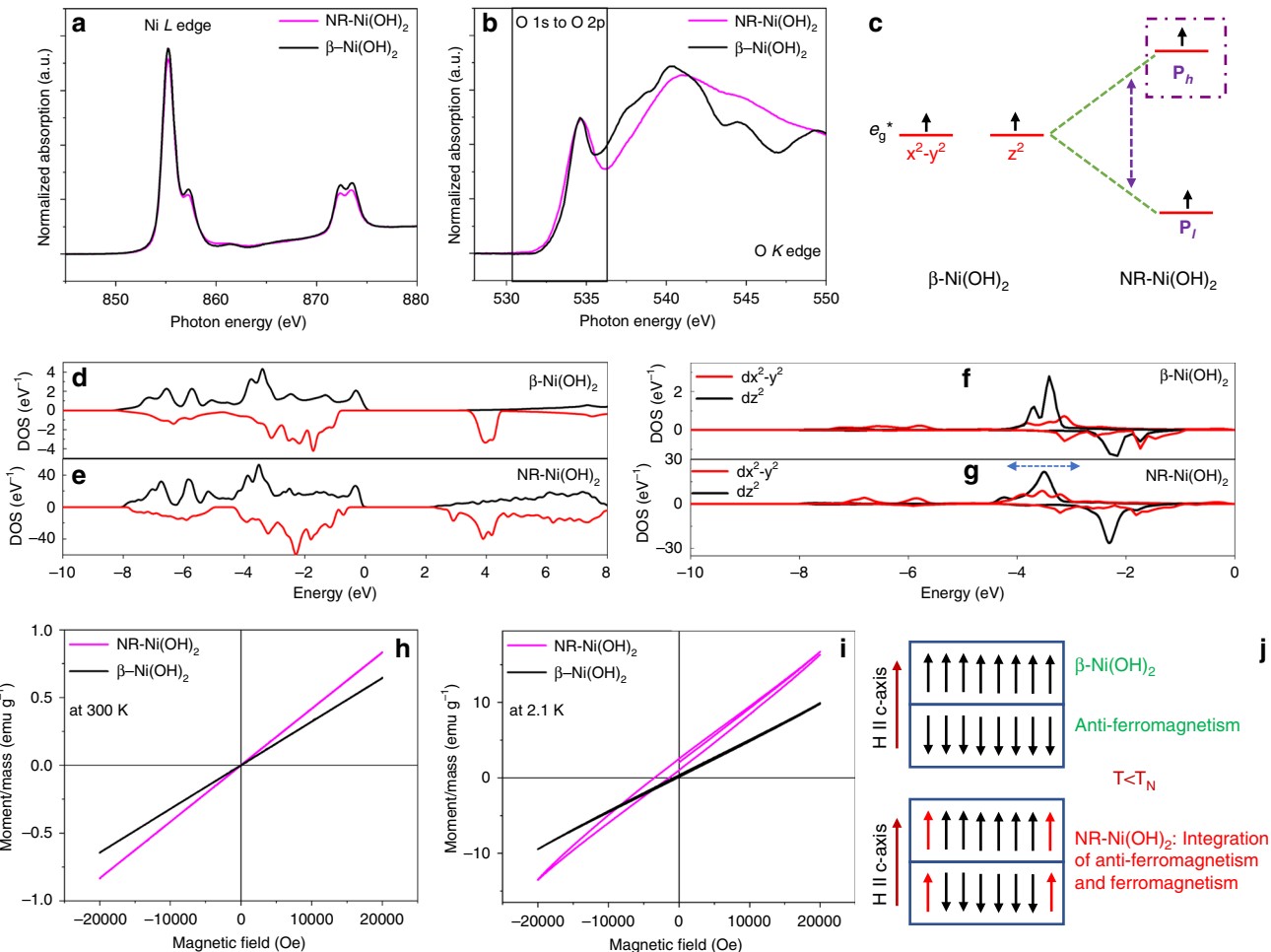

**Fig. 3 Near-edge X-ray absorption fine structure and magnetic measurement. a** Ni $L$ edge spectra of β-Ni(OH)$_2$ and NR-Ni(OH)$_2$; **b** O $K$ edge spectra of β-Ni(OH)$_2$ and NR-Ni(OH)$_2$; **c** schematic of NR-Ni(OH)$_2$ $e_g^\star$ orbital splitting; **d, e** projected density of states (PDOS) of β-Ni(OH)$_2$ and NR-Ni(OH)$_2$, respectively; **f, g** PDOS of $e_g^\star$ orbitals for β-Ni(OH)$_2$ and NR-Ni(OH)$_2$, respectively; **h** $M$–$H$ loops of β-Ni(OH)$_2$ and NR-Ni(OH)$_2$ at 300 K; **i** $M$–$H$ loops of β-Ni(OH)$_2$ and NR-Ni(OH)$_2$ at 2.1 K; **j** schematic of β-Ni(OH)$_2$ and NR-Ni(OH)$_2$ magnetic properties.

transition metal hydroxide usually occurs on the filled lower-Hubbard band (LHB)[21,22]. The detailed electronic orbitals of the traditional Ni(OH)$_2$ are shown in Supplementary Fig. 10. To unveil the factors leading to the superior MOR activity exhibited by the alternating four-six-coordinated NR-Ni(OH)$_2$, the electronic structure is first examined using near-edge X-ray absorption fine structure (NEXAFS). For comparison, conventional β-Ni(OH)$_2$ is used as the benchmark sample. Figure 3a, b shows the Ni $L_{2,3}$ and O $K$ edge X-ray absorption spectra. Ni $L$ edge data of NR-Ni(OH)$_2$ reveals a generally similar profile to that of β-Ni(OH)$_2$, which implies that both samples have very similar local chemical environment. This result suggests that all Ni atoms in NR-Ni(OH)$_2$ are arranged in NiO$_6$ octahedron, which agrees well with our previous report[15]. However, some mismatch exists between NR-Ni(OH)$_2$ and β-Ni(OH)$_2$ as all the peaks are broader for NR-Ni(OH)$_2$. In order to further understand the structural differences, O $K$ edge X-ray absorption is analyzed (Fig. 3b). The energy range between 530.5 eV to 537.5 eV corresponds to O 1 s to O 2p jump, which is mainly related to the electronic states of $e_g^\star$. No pre-peaks are found in this energy range which shows that NR-Ni(OH)$_2$ is well-crystallized[23]. At the same time, the peak located at 534.0 eV becomes broad which can suggest either there are more electronic states in the $e_g^\star$ orbitals, or the splitting of $e_g^\star$ orbitals has occurred. The schematic diagram of $e_g^\star$

splitting is shown in Fig. 3c, with the higher energy orbital denoted as $P_h$ and the lower energy orbital denoted as $P_l$. The energy level splitting becomes more obvious with all the peaks becoming broader in energy range larger than 537.5 eV (O 1 s to higher orbitals jump, e.g., O 3p).

The $e_g^\star$ splitting can be further understood using density functional theory (DFT). Figure 3d, e show the PDOS of β-Ni(OH)$_2$ and NR-Ni(OH)$_2$, respectively. Above the Fermi level (>0 eV), the black peak (3 to 7 eV, $a_{1g}^\star$ orbital) of the NR-Ni(OH)$_2$ is red-shifted as compared with β-Ni(OH)$_2$, and it overlaps with the red peaks (2–5 eV, unoccupied $e_g^\star$ orbitals or upper-Hubbard band (UHB)). Since $a_{1g}^\star$ is a delocalized orbital, its overlapping with the UHB results in the delocalization of the UHB. Furthermore, there is a splitting of peaks (2–5 eV) for NR-Ni(OH)$_2$, which indicates the splitting of the UHB. This theoretical result agrees well with the NEXAS data (Fig. 3b). The isosurface charge density indicates that the UHB delocalization in NR-Ni(OH)$_2$ is resulted from the four-coordinated Ni atoms (Supplementary Fig. 11). The $e_g^\star$ PDOS of β-Ni(OH)$_2$ and NR-Ni(OH)$_2$ are plotted to analyze the LHB electronic states shown in Fig. 3f, g. The result shows the broadening and splitting of $e_g^\star$ orbitals in the energy range −8 to −4 and 2 to 6 eV. This indicates that the LHB of NR-Ni(OH)$_2$ has become delocalized. The $e_g^\star$ PDOS of edge Ni atoms are plotted to analyze the LHB

delocalization in NR-Ni(OH)$_2$ (Fig. 3g). The results indicated that the broadening and splitting of $e_g{}^*$ orbitals resulted from the edge Ni atoms. The isosurface charge density is further calculated to analyze the LHB delocalization in NR-Ni(OH)$_2$ (Supplementary Fig. 12 and Fig. 4c). There is an obvious shape deformation for edge O and Ni atoms in NR-Ni(OH)$_2$. At the same time, the orbitals of these edge atoms become larger as compared with those of interior atoms. These results clearly indicate that the unique electronic states of the four-coordinated Ni atoms are the reasons for LHB delocalization.

The electron delocalization of the $e_g{}^*$ orbitals is further analyzed using a superconducting quantum interface device (SQUID) magnetometer. The Neel temperature ($T_N$) is firstly identified through variations in the moment as a function of temperature for the ZFC (zero-field cooled) and FC (field cooled) cases with applied magnetic field H = 100 Oe. As shown in in Supplementary Fig. 13, $T_N$ for NR-Ni(OH)$_2$ and β-Ni(OH)$_2$ are 21.95 K and 24.75 K, respectively. Both NR-Ni(OH)$_2$ and β-Ni(OH)$_2$ exhibit paramagnetic behavior when the temperature is above $T_N$ (Fig. 3h). However, when the temperature is below $T_N$, β-Ni(OH)$_2$ shows anti-ferromagnetic behavior, while NR-Ni(OH)$_2$ shows an integration of both ferromagnetic and anti-ferromagnetic behavior (Fig. 3i). It is reported that the Ni$^{2+}$ moments in (00$l$) planes are ferromagnetically ordered[24]. However, due to the strong interlayer bonding energy with the neighboring (00$l$), Ni$^{2+}$ moments are antiferromagnetically ordered along the $c$-axis[24]. This means that the electron spin direction on ($x^2$–$y^2$) and $z^2$ orbitals ($e_g{}^*$ energy band) in neighboring (00$l$) planes are inverse. The electronic orbitals of Ni(OH)$_2$ are provided in Supplementary Fig. 10a. Thus, the remanence ($M_r$) depends on the number of uncompensated surface layers. For an even number of (00$l$) layers, the moments are compensated which leads to anti-ferromagnetic ordering (Fig. 3j). On the other hand, the uncompensated moments in an odd number of (00$l$) layers will yield $M_r$ which ultimately leads to ferrimagnetic ordering. As a result, the traditional Ni(OH)$_2$ can either exhibit anti-ferromagnetism or ferromagnetism, depending on the number of layers.

The low dose scanning transmission electron microscopy (STEM) indicates that more than 89% of NR-Ni(OH)$_2$ are double-layer in our previous work[15]. Based on the statistical orientation of NR-Ni(OH)$_2$, only 1/3 will be oriented with H‖c axis on average. As a result, only 3% of NR-Ni(OH)$_2$ (since these 3% NR-Ni(OH)$_2$ consist of three layers and the two layers are compensated, which results in only 1% uncompensated layers contributing to the effect) would yield $M_r$ leading to ferrimagnetic ordering. Due to the nearly negligible amount of uncompensated layers in the NR-Ni(OH)$_2$, their effect could be practically ignored. Based on the theory mentioned earlier, NR-Ni(OH)$_2$ should exhibit anti-ferromagnetism due to the mostly double layer structure. However, our experiment result shows that NR-Ni(OH)$_2$ exhibits both ferromagnetism and anti-ferromagnetism, which is in stark contrast to the earlier theory. Furthermore, based on the above discussion, this ferromagnetic property should not originate from the nearly negligible amount of uncompensated surface layers. Thus, this unique phenomenon can be attributed to the presence of four-coordinated Ni atoms, as their electron spins in the $e_g{}^*$ energy band are no longer confined by the interlayer bonding energy. As a result, the electron spin directions in the $e_g{}^*$ orbitals with neighboring (00$l$) planes are the same, leading to ferrimagnetic ordering (Fig. 3j). This means that the electrons become delocalized over the four-coordinated Ni atom. Hence, based on the NEXAFS and DFT simulation results, these delocalized electrons are located at the LHB higher orbital ($P_h$ orbital), which possess higher energy and are near to the Fermi level.

It is well known that the extension of octahedron structure would lead to $e_g{}^*$ level splitting. When this extension reaches infinity, the octahedron structure will transform into square planar with ($x^2$–$y^2$) orbital energy larger than the Fermi energy level (Fig. 4a). As compared to the traditional six-coordinated NiO$_6$ octahedron, NR-Ni(OH)$_2$ possesses four-coordinated Ni atoms with two dangling bonds. Instead of the typical formation of a NiO$_4$ square planar configuration for a four-coordinated Ni atom, the four-coordinated Ni atom in NR-Ni(OH)$_2$ is arranged in a NiO$_6$ octahedron configuration. Thus, this structure could be perceived as an intermediate configuration between an extended octahedron and square planar as shown in Fig. 4a. As such, the four-coordinated Ni atoms arrangement is considered to be a special distorted octahedron with delocalized electron states.

After identifying the delocalization of electrons in the four-coordinated Ni atom in NR-Ni(OH)$_2$, the next step is to investigate the effect of these delocalized electrons on the MOR performance. To reveal some insights to the electron transfer mechanism in our four-six-coordinated NR-Ni(OH)$_2$, an illustration is shown in Fig. 4b. The delocalized electrons in the highest electron occupied orbital ($P_h$) of the four-coordinated Ni atom are weakly or not bonded. These delocalized electron orbitals can serve as charge-transfer sites, which transfer electrons from $P_h$ to ($x^2$-y$^2$) and $z^2$ orbitals of its neighboring six-coordinated Ni atom. Eventually, these electrons will then be transferred to the ($P_h$) of the next neighboring four-coordinated Ni atom (Fig. 4b), and such process continues until the electrons enter into the current collector. This proposed electron transfer mechanism agrees well with the isosurface charge density of NR-Ni(OH)$_2$ LHB result as shown in Fig. 4c. The isosurface charge distribution indicates the overlapping of the edge O atoms with the adjacent four Ni atoms, which could form an effective charge-transfer pathway as the black arrows shows. For square planar structure whose Ni atoms are four-coordinated, the orbitals below Fermi energy level are fully occupied which hinders the transfer of electrons into adjacent orbitals (Fig. 4a). This explains why the four-coordinated Ni atom must be paired to a neighboring six-coordinated Ni atom in order to achieve efficient charge transfer.

However, for the traditional Ni(OH)$_2$, $e_g{}^*$ orbitals in (M–O)* antibonding bands are occupied by two spin-up electrons which hinders the transfer of electrons into the adjacent $e_g{}^*$ orbital due to the Pauli exclusion principle. When Ni(OH)$_2$ oxidizes into NiOOH, there would only be one spin up electron in the $e_g{}^*$ orbital, which allows effective electron transfer into the adjacent $e_g{}^*$ orbital (Supplementary Fig. 14). This is in good agreement with the reported MOR process of traditional Ni(OH)$_2$, which can be summarized as follows; (a) nickel oxidation of Ni$^{2+}$ to Ni$^{3+}$, and (b) methanol oxidation Ni$^{3+}$ + CH$_3$OH → Ni$^{2+}$ + products (spontaneous process)[12]. This indicates that Ni(OH)$_2$ is unable to catalyze methanol oxidation without being oxidized to NiOOH. As the potential for Ni oxidation is usually larger than 1.35 V vs. RHE, traditional Ni(OH)$_2$ cannot trigger the operation of DMFCs. This agrees well with our experimental results (Supplementary Table 1). Based on these discussions, we can conclude that achieving efficient charge transfer is the crucial factor in realizing methanol oxidation. We have demonstrated that the electron charge transfer in our periodically arranged four-six-coordinated Ni atoms in NR-Ni(OH)$_2$ is indeed efficient, which ultimately contributes toward efficient MOR. It was reported that there was no formation of NR-Ni(OH)$_2$ with alternating four-six coordinated Ni atoms when reacting Ni(OH)$_2$ with S[25]. Combining the results obtained in this work, it is realised that the oxygen content and Ni–O bond in NiS$_2$ may exert certain influence on the formation of NR-Ni(OH)$_2$, which requires further comprehensive studies.

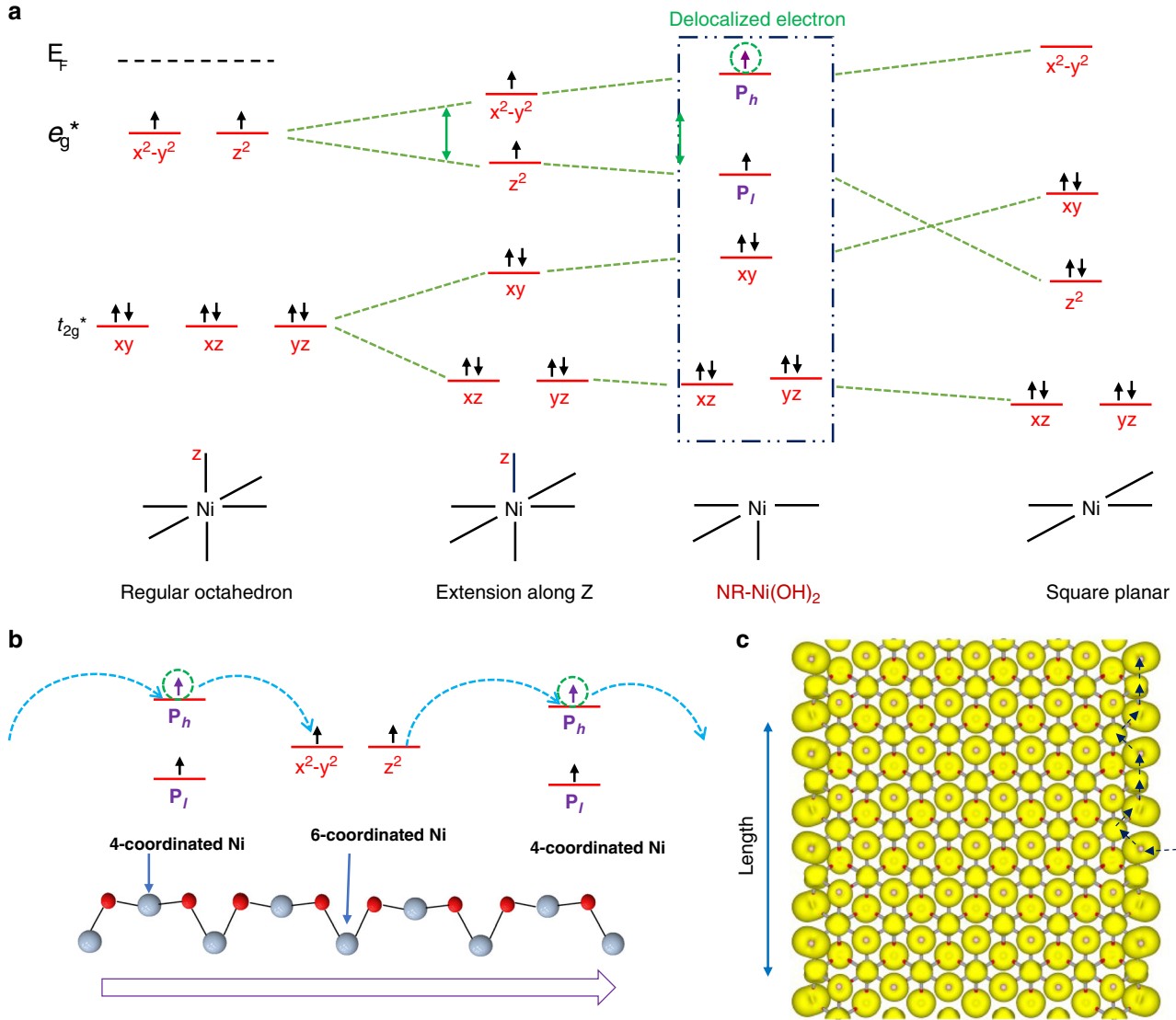

**Fig. 4 Schematic of NR-Ni(OH)$_2$ electronic orbitals and charge transfer pathway. a** $t_{2g}{}^\star$ and $e_g{}^\star$ electronic orbitals of regular NiO$_6$ octahedron, NiO$_6$ octahedron extended along $z$ axis, NR-Ni(OH)$_2$, and NiO$_4$ square planar; **b** charge transfer process on alternating four-six-coordinated arrangement in NR-Ni(OH)$_2$; **c** isosurface of charge density of NR-Ni(OH)$_2$ LHB.

## Discussion

In summary, our previously reported nickel hydroxide nanoribbon structure (NR-Ni(OH)$_2$) is able to show remarkable activity with an onset potential of 0.55 V vs. RHE. This is mainly due to the generation of available charge-transfer orbitals for electrons delocalized near the Fermi energy level using a four-coordinated Ni atom, which significantly enhances the six-electron MOR transfer process. Thus, due to our periodically arranged four-six-coordinated Ni atoms NR-Ni(OH)$_2$, methanol conversion to CO$_2$ is highly efficient. Our initial attempt to construct a NR-Ni(OH)$_2$ assembled DFMC device shows an operable OCV of 0.58 V, which indicates its feasibility for DMFC operation. Furthermore, the preliminary investigation of the as-assembled DFMC demonstrates durable operation with zero CO poisoning. With these positive demonstrations, it is expected that this strategy will offer a practical gateway toward the development of high performance and durable non-PGM DMFC.

## Methods

**Electrode preparation**. In brief, the catalyst was ground thoroughly with poly-vinylidene fluoride (PVDF) as the binder and carbon black as the conductive

medium with a mass ratio: 7:1:2. The solvent used for the grinding process was N-methy-2-pyrrolidone (NMP) solution. Eventually, a uniform slurry was obtained. The uniform slurry was then pasted onto the carbon paper (0.18 mm, with 77% porosity). The carbon paper with coated slurry was heated in a conventional oven overnight at 80 °C.

## Characterization

*Near-edge X-ray absorption fine structure*. Near-edge X-ray absorption fine structure (NEXAFS) spectra were measured at Singapore Synchrotron Light Sources (SSLS). The Ni $L_2$, $L_3$ and O K edge NEXAFS spectra was collected in total electron yield mode with a photon energy resolution of 350 meV. The photon energy was calibrated using the characteristic intensity dip at 284.4 eV from the contamination carbon of the beamline optical components. All NEXAFS spectra are normalized to the incident photon intensity ($I_0$) monitored by the focusing mirror.

*Fourier transform infrared spectra*. Fourier transform infrared (FTIR) spectra were collected with a PerkinElmer Frontier MIR/FIR system by 16 scans with a nominal resolution of 1 cm$^{-1}$.

*Electrochemical measurement*. Electrochemical measurements were performed using a three-electrode system connected to an electrochemical workstation (VMP3, Bio-logic Inc) with built-in electrochemical impedance spectroscopy (EIS) analyzer. The reference electrode is a Hg/HgO electrode, which was calibrated through the method below.

*Magnetic measurement.* The magnetic properties were investigated using a superconducting quantum interface device (SQUID) magnetometer.

## Direct methanol fuel cell assembly and performance evaluation

*Ion conductor solution preparation.* 0.25 g of Fumasep FAA-3-PK-130 alkaline membrane was firstly cut into small pieces. Then, the small membrane pieces were dissolved in 5 ml N-methy-2-pyrrolidone (NMP) solution through heating at 150 °C for 3 h, to give a brown ion conductor solution.

*Membrane electrode assembly (MEA) preparation.* The catalyst solution was prepared by mixing 40% catalyst, 40% carbon black, 30% ion conductor solution. Then, the ink was coated on gas diffusion layer at 60 °C. Next, the MEA was prepared by pressing the anode, membrane and cathode electrode under 1.4 MPa at 80 °C for 3 min, with the size of 5 cm$^2$.

*DMFC performance evaluation.* DMFC devices was assemble as shown in Fig. S4, and the performance was evaluated using a fuel cell test system (850e, Scribner) at 80 °C, with a 1 M methanol concentration and O$_2$ flow rate of 190 cc/min. The active cell area of DMFC was 5 cm$^2$.

## Computational method

*NR-Ni(OH)$_2$ construction.* All calculations were carried out using the density functional theory (DFT) with the generalized Perdew-Burke-Ernzerhof (PBE) and DFT + U ($U_{eff} = U - J = 5.3$), and the projector augmented-wave (PAW) pseudopotential planewave method as implemented in the VASP code. For the PAW pseudopotential, we included $3d^84s^2$, $2s^22p^4$ and $1s^1$ were treated as valence electrons for Ni, O, and H atoms, respectively. A $12 \times 12 \times 10$ Monkhorst-Pack (MP) k-point grid was used for Ni(OH)$_2$ unitcell geometry optimization calculations. Good convergence was obtained with these parameters, and the total energy was converged to $1 \times 10^{-6}$ eV per atom. Energy convergence with respect to the plane wave cutoff was tested by varying this setting between 300 and 600 eV considering the electron spinpolarization. Convergence to within 10 meV was achieved with a cutoff energy of 500 eV for Ni(OH)$_2$ unitcell. We carried out calculations with the van der Waals (vdW) correction by employing optPBE-vdW functional, and the reaction paths of sequential hydrogen abstraction steps are calculated by means of the Nudged Elastic Band (NEB) method. Nickel hydroxide has two different crystalline forms, named α- and β-Ni(OH)$_2$, and latter is the most stable form of nickel hydroxide with the point group of $P\bar{3}m1$ (No. 164). β-Ni(OH)$_2$ has a layered structure, with OH$^-$ anions between nickel ions, and layers stack together by hydrogen bonds. In our calculations, we optimized the unitcell of Ni(OH)$_2$ and the calculated lattice constants are, a = b = 3.04 and c = 4.61 Å. Compared with the experimental values of a = b = 3.12 Å and c = 4.66 Å[26], we consider that our basic settings are correct and our calculation results are reliable. Based on the optimized layered unitcell of Ni(OH)$_2$ and the experimental results, we constructed Ni(OH)$_2$ nanoribbons (NRs) supercell for methanol decomposition study (Supplementary Fig. 6). It should be noted that all energies reported here are corrected with the zero-point energy (ZPE).

*Calculations of MOR activity.* We carried out calculations with the van der Waals (vdW) correction by employing the optPBE-vdW functional, and the reaction paths of sequential hydrogen abstraction steps are calculated by means of the NEB method. It should be noted that all energies reported here are corrected with the ZPE.

## Data availability

The authors declare that all data supporting the finding of this study are available within the paper and its Supplementary information files.

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

## Acknowledgements

This work is financially supported by Singapore MOE Tier 1 R284000226114 and MOE Tier 2 (MOE2018-T2-1-149), Technology and Research (A*STAR) of Singapore. This research is also supported by A*STAR with a Grant No. of 152-70-00017 and computational resources were provided by National Supercomputing Centre Singapore (NSCC) and A*STAR Computational Resource Centre, Singapore (A*CRC). This project was partly supported by the Science and Engineering Research Council (SERC) of A*STAR of Singapore. S. J. Pennycook is grateful to the National University of Singapore and a MOE Tier 2 (MOE2017-T2-1-129) project for support. W.C. Hao is grateful to the National Science Foundation of China (No. 11874003 and No.51672018). X.P. Wang is grateful to graphic designer Hui Wang for her help in figure design.

## Author contributions

X.P.W., S.B.X., and J.M.X. conceived the idea. X.P.W. performed synthesis and electrochemical measurement of the samples. W.S.V.L., H.J.W., H.W., and S.J.P. were responsible for the analysis of STEM and FTIR results. P.C., L.Z., X.S.Z., and Z.B.W. were in charge of DMFC test. S.B.X., A.B., C.Z.D., and Y.H.D. were responsible for the NEXAS characterization. Z.G.Y., and P.R.H. carried out DFT simulations. W.C.H. conducted magnetic measurement. J.M.X. is in charge of the overall project and preparation of the paper.

## Competing interests

The authors declare no competing interests.
