## [Peer Review File · Nature Communications]

REVIEWER COMMENTS

Reviewer #1 (Remarks to the Author):

Due to the increasing demands of new generation form energies, the increasing methanol availability and managing is a challenge. Thus, the development an efficient device that can effectively convert methanol into electrical power is by no means a discussion. In this sense, the results presented by Wang et al. in this study, are of significant importance to reduce the reliance in PGMs by developing alternative nanostructured materials using durable and reasonable low cost-effective catalyst that trigger the MOR reaction in DFMC technologies.

The manuscript is well written, settling well the problem and background, the materials characterization techniques portfolio are well suited for the project, nice match among the experimental previous and actual results vs DFT calculations, and the results are logically presented and properly discussed.

This referee prefers to see a concise section describing the DFT calculation details on Methods section. Thus, in Methods section (if not, then in the part of SI file), it should be included the basic data for these calculations being reproduced on the same or similar systems, e.g., what box- software do you used, the energy cuts applied, etc.

Some minor suggestions and typos to be corrected.

a. Seems to me that there is a typo in the data on p. 3, lines 5, it says 0.4-6 V, is not 0.4-0.6 V? Please review.

b. There are two typos in caption of Fig. S1. There should be read "traditional", and "dissolved".

c. This reviewer considere there are difference in the meaning of "calculations" (what you did) vs "simulations". Kindly, evaluate to change the use of these words since they strictly mean not the same.

This reviewer does not have inconvenient to accept for publish the paper, perfoming these mentioned minor corrections.

Reviewer #2 (Remarks to the Author):

The authors illustrated the potential of a periodically arranged four-six-coordinated nickel hydroxide nanoribbon structure (NR-Ni(OH)₂) in effectively catalyzing the MOR reaction. The remarkable performance of the electrode was attributed to the formation of charge transfer orbitals through delocalization of electrons near the fermi energy level. Detailed characterization and theoretical calculations have been given in support of the claim.

Strain stabilized nickel hydroxide nanoribbons were synthesized following the authors previous work, by a 4-step route. First Ni was electrodeposited and converted to NiS₂ upon desulfurization which was electrochemically oxidized to NiOOH forming Ni(OH)₂ as an intermediate, the NiOOH from the 3rd step was then reduced back to Ni(OH)₂ through ethanol reduction.

Elaborate explanation of the effect or contribution of each synthesis step on the stabilization of non-stoichiometric Ni(OH)₂ with 4 coordinated Ni would add to the significance of the study and pave the way as a methodology for systematic formation and control of low coordinated and non-stoichiometric transition metal hydroxides. In this regard, providing answers to the following questions is deemed necessary.

1. What is the effect or contribution of each synthesis step on the stabilization of low coordination Ni in Ni(OH)₂,
2. What is the difference between the Ni(OH)₂ at step 3 (obtained during electrochemical oxidation) before complete oxidation to NiOOH and the Ni(OH)₂ obtained after ethanol

reduction of the NiOOH obtained in step 3?

3. What is the relevance of NiS₂ formation in the synthesis procedure?

Reviewer #3 (Remarks to the Author):

The authors describe a special nanostructured Nickel Hydroxide catalyst for the electro-catalytic oxidation of methanol. The difference between their Nanoribbon-structured Nickel Hydroxide and other Nickel Hydroxide structures is studied in detail by measurements and simulations in order to elucidate the improved activity for methanol oxidation. With these methods, the authors succeed to attribute the catalytic activity to the electronic structure of the material.

The material has been studied previously by the authors for water splitting, as cited properly in the manuscript. The study on methanol oxidation is novel and a significant advancement to justify publication in nature communications.

In my opinion, the paper should be published with a few minor revisions:

p.1, l.23: The references are inappropriate to show the general advantages of methanol as a fuel.

p.2. l.8: Low durability of PGMs in methanol oxidation is not true. Lifetimes of several thousands of hours have been demonstrated by several groups in acidic membrane based DMFCs. Products are on the market.

p.4. l.8 and other positions: The authors should comment on the effect of CO₂ as the main product of methanol oxidation on the conductivity of the membrane in alkaline environment. I would expect an exchange of OH⁻ with CO₃²⁻ with a significant reduction in conductivity.

p.6 l.15: how is stability defined here?

p.16 l.25: The size of the active cell area must be given; otherwise, the flow rate of 190 cc/min has no meaning.

The supplemental material is extremely extensive and some of the figures could easily be removed: Figure S4 and S5 (top) are more or less the same and should be widely available in the literature, Figure S20b is the same as Figure 4c. Figure S22 is from the previous publication of the authors and Figure S23 is textbook knowledge.

Editorial hints:

p.2 l.5: 0.4-0.6 V instead of 0.4-6 V (which would mean 0.4 – 6.0V)

p.8 l.5: Please check the wavenumbers shown in Fig. 2c

Fig. S16 b and d: x-Axis should be log (f)

Fig.3: g is missing in the consecutive numbering of sub-figures

p.13 l.5: Fig 3k (instead of Fig.2k)

p.16 l.26: The reference to Fig.4a and S31 makes no sense here. Fig. S4 and S5?

Andreas Glösen

REVIEWER COMMENTS

Reviewer #1 (Remarks to the Author):

Due to the increasing demands of new generation form energies, the increasing methanol availability and managing is a challenge. Thus, the development an efficient device that can effectively convert methanol into electrical power is by no means a discussion. In this sense, the results presented by Wang et al. in this study, are of significant importance to reduce the reliance in PGMs by developing alternative nanostructured materials using durable and reasonable low cost-effective catalyst that trigger the MOR reaction in DFMC technologies.

The manuscript is well written, settling well the problem and background, the materials characterization techniques portfolio are well suited for the project, nice match among the experimental previous and actual results vs DFT calculations, and the results are logically presented and properly discussed.

This referee prefers to see a concise section describing the DFT calculation details on Methods section. Thus, in Methods section (if not, then in the part of SI file), it should be included the basic data for these calculations being reproduced on the same or similar systems, e.g., what box-software do you used, the energy cuts applied, etc.

Response: We are grateful for the positive endorsement by the referee. We have included the computational method in the revised SI in Page 9.

All calculations were carried out using the density functional theory (DFT) with the generalized Perdew-Burke-Ernzerhof (PBE) and DFT+ U ($U_{\text{eff}}=U-J=5.3$) (7), and the projector augmented-wave (PAW) pseudopotential planewave method, as implemented in the VASP code (8, 9). For the PAW pseudopotential, $3d^84s^2$, $2s^22p^4$, and $1s^1$ were treated as the valence electrons for Ni, O, and H atoms, respectively. A $12\times 12\times 10$ Monkhorst-Pack (MP) k-point grid was used for Ni(OH)₂ unit cell geometry optimization calculations. Good convergence was obtained with these parameters, and the total energy was converged to 1×10^{-6} eV per atom. Energy convergence with respect to the plane wave cutoff was tested by varying this setting between 300 and 600 eV considering the electron spin polarization. Convergence to within 10 meV was achieved with a cutoff energy of 500 eV for Ni(OH)₂ unit cell. We carried out the calculations with the van der Waals (vdW) correction by employing optPBE-vdW functional (10), and the reaction paths of sequential hydrogen abstraction steps are calculated by means of the Nudged Elastic Band (NEB) method (11). Nickel hydroxide has two different crystalline forms, *i.e.*, α - and β -Ni(OH)₂, and the latter is the most stable form of nickel hydroxide with the point group of $P\bar{3}m1$ (No. 164). β -Ni(OH)₂ has a layered structure, with OH⁻ anions between the nickel ions, and the layers are stacked together *via* hydrogen bonds. In our calculations, we optimized the unit cell of Ni(OH)₂ and the calculated lattice constants are $a=b=3.04$ and $c=4.61$ Å. Compared with the experimental values of $a=b=3.12$ Å and $c=4.66$ Å (12), our basic settings are accurate and our calculation results are considered to be reliable. Based the optimized layered unit cell of Ni(OH)₂ and the experimental results, we constructed Ni(OH)₂ nanoribbons (NRs)

supercell shown in Fig. S6 for methanol decomposition study. It should be noted that all energies reported here are corrected with the zero-point energy (ZPE).

Some minor suggestions and typos to be corrected.

a. Seems to me that there is a typo in the data on p. 3, lines 5, it says 0.4-6 V, is not 0.4-0.6 V? Please review.

Response: We are thankful to the referee for pointing out this error. We have revised 0.4-6V to 0.4-0.6V in the revised manuscript Page 3, line 5.

b. There are two typos in caption of Fig. S1. There should be read "traditional", and "dissolved".

Response: We are thankful to the referee for pointing out this error. We have revised these two typos in caption of Fig. S1 to "traditional" and "dissolved" in our revised SI.

c. This reviewer considers there are differences in the meaning of "calculations" (what you did) vs "simulations". Kindly, evaluate to change the use of these words since they strictly mean not the same.

Response: We appreciate this constructive suggestion. We have heeded the referee's suggestion to consider the difference in the meaning of "calculations" and "simulation". For all the simulation conducted in this work, we have changed the term to "calculation" instead of "simulation".

This reviewer does not have any inconvenience to accept for publication of the paper, performing these mentioned minor corrections.

Reviewer #2 (Remarks to the Author):

The authors illustrated the potential of a periodically arranged four-six-coordinated nickel hydroxide nanoribbon structure (NR-Ni(OH)₂) in effectively catalyzing the MOR reaction. The remarkable performance of the electrode was attributed to the formation of charge transfer orbitals through delocalization of electrons near the Fermi energy level. Detailed characterization and theoretical calculations have been given in support of the claim.

Strain-stabilized nickel hydroxide nanoribbons were synthesized following the authors' previous work, by a 4-step route. First Ni was electrodeposited and converted to NiS₂ upon desulfurization, which was electrochemically oxidized to NiOOH, forming Ni(OH)₂ as an intermediate. The NiOOH from the 3rd step was then reduced back to Ni(OH)₂ through ethanol reduction.

Elaborate explanation of the effect or contribution of each synthesis step on the stabilization of non-stoichiometric Ni(OH)₂ with 4-coordinated Ni would add to the significance of the study and pave the way as a methodology for systematic formation and control of low-coordinated and non-stoichiometric transition metal hydroxides. In this regard, providing answers to the following questions is deemed necessary.

1. What is the effect or contribution of each synthesis step on the stabilization of low coordination Ni in Ni(OH)₂.
2. What is the difference between the Ni(OH)₂ at step 3 (obtained during electrochemical oxidation) before complete oxidation to NiOOH and the Ni(OH)₂ obtained after ethanol reduction of the NiOOH obtained in step 3?
3. What is the relevance of NiS₂ formation in the synthesis procedure? The NiS₂ is very important.

Response:

Since these three questions are all related to the preparation of NR-Ni(OH)₂, we have combined our responses to these questions in this section. As the reviewer has pointed, NR-Ni(OH)₂ was prepared through a multi-step process, which include (1) formation of NiS₂, (2) electro-oxidation of NiS₂, and (3) ethanol reduction of NR-NiOOH. Among these three steps, both the synthesis of NiS₂ precursor and its electrooxidation are the key steps in the formation of NR-Ni(OH)₂. On the other hand, based on the study, ethanol reduction process does not affect the unique nanoribbon structure. Here we will discuss each of the steps involved in the formation of NR-Ni(OH)₂.

(1) Electro-oxidation of NiS₂

In our previous work (*Energy Environ. Sci.*, 2020, 13, 229), we have employed the use of *operando* XAFS conducted at a fixed current of 10 mA cm⁻² to study the entire electro-oxidation process. Based on our result, it is revealed that there is a phase transition from NiS₂ to NR-Ni(OH)₂, and finally to NR-NiOOH. This is the key step for the formation of NR-Ni(OH)₂.

(2) Synthesis of NiS₂ precursor

To address question 3 posed by the reviewer, the synthesis of NiS₂ and its importance are discussed. As the electro-oxidation process of NiS₂ is the key step in the preparation of NR-Ni(OH)₂, it is essential to understand the effect of NiS₂ precursor on the formation of NR-Ni(OH)₂. In order to investigate this, another NiS₂ was prepared by reacting Ni(OH)₂ with S at 400 °C in the presence of Ar atmosphere. (*ChemElectroChem* 2019, 6, 2741-2747). The sample was analysed using XRD (Fig. R2.1), which confirmed the successful preparation of NiS₂, denoted as H-NiS₂. As shown in Fig. R2.2, its morphology was relatively similar to that of the NiS₂ prepared by reacting metal Ni and S, denoted as M-NiS₂.

Fig. R2.1. The XRD spectrum of H-NiS₂.

Fig. R2.2. The SEM images of (a) H-NiS₂ and (b) M-NiS₂.

However, when electro-oxidation was conducted on both samples at the current of 10 mA cm⁻² for 10 hours, distinct difference was observed. As shown in Fig. R2.3, there was no obvious potential change in the electro-oxidation curve of H-NiS₂, while there were four potential stages for M-NiS₂. Moreover, the morphology of Ni(OH)₂ obtained from H-NiS₂ was round shaped with size of 2-10 nm (Fig. R2.4), which was greatly different from the observed nanoribbon structures in the sample obtained through the electro-oxidation of M-NiS₂. Therefore, it is believed that H-NiS₂ is not suitable in the synthesis of NR-Ni(OH)₂. This indicates that NiS₂ precursor is considered as a key step in the formation of NR-Ni(OH)₂.

Fig. R2.3. The electro-oxidation curves of (a) H-NiS₂ and (b) M-NiS₂.

Fig. R2.4. High-resolution TEM image of sample obtained *via* the electro-oxidation of H-NiS₂. The lattice spacing of 0.23 nm (inset image) corresponds to the (111) plane of NiO.

Thus, based on the results, we believe that there are some differences between M-NiS₂ and H-NiS₂, despite their similar XRD profiles. For instance, one possible difference between these two materials

is the oxygen content. For M-NiS₂, Ni metal was directly reacted with S under Ar, which inevitably resulted in a low oxygen content in the sample. However, for H-NiS₂, it was prepared by reacting Ni(OH)₂ with S. Since Ni-O is a strong chemical bond, it is difficult to be broken down at 400°C completely. This would lead to the remaining partial Ni-O chemical bonds left in the NiS₂, which may affect the formation of NR-Ni(OH)₂. The effect of Ni-O on the formation of NR-Ni(OH)₂ will be further studied in our future work.

(3) Ethanol reduction

This step is related to the second question posed by the reviewer. In order to understand whether there is significant change in the crystal structure of NR-Ni(OH)₂ after reducing NR-NiOOH with ethanol, NR-Ni(OH)₂ was re-oxidized under current density of 10 mA cm⁻². Subsequently, the sample was analysed using XAFS (Fig. R2.5, derived from Fig. S8 in our previous work (*Energy Environ. Sci.*, 2020, 13, 229)). It is observed that these two curves overlapped completely, which indicated that **there was no structural change in NR-Ni(OH)₂ after reducing NR-NiOOH using ethanol**. The reversible phase transition between NR-Ni(OH)₂ and NR-NiOOH agrees well with Ni(OH)₂ alcohol oxidation reaction mechanism, that Ni(OH)₂ was firstly oxidized into NiOOH and NiOOH could be used to oxidize alcohol, then itself would be reduced to Ni(OH)₂ (1, 2).

Fig. R2.5. XAFS of NR-NiOOH, NR-Ni(OH)₂, and re-oxidized NR-Ni(OH)₂ sample. This figure was originally labelled as Fig. S8 in our previous work (*Energy Environ. Sci.*, 2020, 13, 229).

Based on the above discussion, the key steps in the formation of NR-Ni(OH)₂ are (1) synthesis of NiS₂ precursor, and (2) the electro-oxidation of NiS₂. On the other hand, ethanol reduction process did not affect the structure, as shown by the XAFS results. The effect of each synthesis step on the formation of NR-Ni(OH)₂ will be further studied in our future work.

References:

1. D. F. Wu, W. Zhang, D. J. Cheng, Facile synthesis of Cu/NiCu electrocatalysts integrating alloy, core-shell, and one-dimensional structures for efficient methanol oxidation reaction. *ACS. Appl. Mater. Interfaces* 2017, **9**, 19843-19851.
2. S. N. Sun, Y. Zhou, B. L. Hu, Q. C. Zhang, Z. C. J. Xu, Ethylene glycol and ethanol oxidation on spinel Ni-Co oxides in alkaline. *Journal of The Electrochemical Society*, **2016**, 163 (2) H99-H104.

Reviewer #3 (Remarks to the Author):

The authors describe a special nanostructured Nickel Hydroxide catalyst for the electro-catalytic oxidation of methanol. The difference between their Nanoribbon-structured Nickel Hydroxide and other Nickel Hydroxide structures is studied in detail by measurements and simulations in order to elucidate the improved activity for methanol oxidation. With these methods, the authors succeed to attribute the catalytic activity to the electronic structure of the material.

The material has been studied previously by the authors for water splitting, as cited properly in the manuscript. The study on methanol oxidation is novel and a significant advancement to justify publication in nature communications.

In my opinion, the paper should be published with a few minor revisions:

p.1, 1.23: The references are inappropriate to show the general advantages of methanol as a fuel.

Response: We appreciate the constructive comment. To highlight the advantages of methanol as fuel, we have included more relevant references in the revised manuscript. The following references are included in the revised manuscript:

- 1), M. Müller, N. Kimiaie, A. Gißen, D. Stolten, The long way of achieving a durability of 20,000 H I a DMFC system. *Adv. Sci. Technol.* 2014, 93, 56-60;
- 2), A. Gißen, F. Dionigi, P. Paciok, M. Heggen, M. Müller, L. Gan, P. Strasser, R. E. Dunin-Borkowski, D. Stolten, Dealloyed PtNi-Core-shell nanocatalysts enable significant lowering of Pt electrode content in direct methanol fuel cell. *ACS Catal.* 2019, 9, 3764-3772;
- 3), J. Mergel, H. Janssen, M. Müller, J. Wilhelm, D. Stolten, Development of direct methanol fuel cell systems for material handling applications. *J. Fuel Cell Sci. Technol.* 2012, 9, 0311;
- 4), M. Müller, N. Kimiaie, A. Gißen, Direct methanol fuel cell systems for backup power-influence of the standby procedure on the lifetime. *Int. J. Hydrogen Energy* 2014, 39, 21739-12745.

p.2. 1.8: Low durability of PGMs in methanol oxidation is not true. Lifetimes of several thousands of hours have been demonstrated by several groups in acidic membrane based DMFCs. Products are on the market.

Response: We sincerely appreciate the insight provided by the reviewer. To accurately depict the disadvantage of PGMs in methanol oxidation, we have removed the portion stating the poor durability of PGMs as one of the factors hindering their practicality. Instead, we will only focus on the cost as the most significant disadvantage of PGMs in the revised manuscript, as shown below. This is included as Line 7, Page 3 in the revised manuscript.

While PGMs have been used in DMFCs, their practicality as functional MOR catalyst is stifled by their high cost (3, 8, 11).

p.4. 1.8 and other positions: The authors should comment on the effect of CO₂ as the main product of methanol oxidation on the conductivity of the membrane in alkaline environment. I would expect an exchange of OH⁻ with CO₃²⁻ with a significant reduction in conductivity.

Response: We are grateful to the suggestion and we agree with the reviewer. We have included the discussion on the effect of the exchange of OH⁻ with CO₃²⁻ and HCO₃⁻ on the anion exchange membrane in DMFC, according to the reviewer's advice. The following writeup is included in line 29, Page 6 of the revised manuscript.

The slight decrease in output voltage of DMFC with the use of NR-Ni(OH)₂ may be due to the exchange of OH⁻ with carbonate (CO₃²⁻) and bicarbonate (HCO₃⁻), which can greatly decrease the conductivity of membrane (16).

p.6 1.15: how is stability defined here?

Response: The stability is generally defined as the retained activity of electrocatalyst after long-term operation. In our work, the stability refers to the retained MOR activity after 3600s chronoamperometric operation. The less activity loss, the better stability. In order to increase the clarity of the manuscript, we added the definition of the stability in our revised manuscript (Line 15, Page 6).

Moreover, the stability indicating the retained activity of electrocatalysts after long-term operation, was measured through chronoamperometric measurement using the previous method (13). Over 3600s operation NR-Ni(OH)₂ exhibits 16% in activity, while Pt/C only has 4% activity, suggesting that NR-Ni(OH)₂ has better stability than Pt/C (Fig. S3).

p.16 1.25: The size of the active cell area must be given; otherwise, the flow rate of 190 cc/min has no meaning.

Response: We agree with the reviewer. We have added the size of the active cell area, *i.e.*, 5 cm², in the methods section of our revised manuscript (Line 28, page 16):

The active cell area of the DMFC was 5cm².

The supplemental material is extremely extensive and some of the figures could easily be removed: Figure S4 and S5 (top) are more or less the same and should be widely available in the literature;

Response: We are thankful for the suggestion. In order to make our supplementary material more concise, we have removed Fig. S4 in the revised SI.

Figure S20b is the same as Figure 4c. Figure S22 is from the previous publication of the authors and Figure S23 is textbook knowledge.

Response: We are grateful for the suggestion. We have removed Fig. S20b, Fig. S22, and Fig. 23 to achieve a more concise supplementary material.

Editorial hints:

p.2 1.5: 0.4-0.6 V instead of 0.4-6 V (which would mean 0.4 – 6.0V).

Response: We have revised 0.4-6V to 0.4-0.6V in our revised manuscript (Line 5, Page 3).

p.8 1.5: Please check the wavenumbers shown in Fig. 2c

Response: We have corrected the wave number ranges (900 to 1070 cm⁻¹) in the caption for Fig. 2c of our revised manuscript.

Fig. S16 b and d: x-Axis should be log (f)

Response: We have revised the x-Axis of Fig. S16 b and d to log (f) in our revised SI.

Fig.3: g is missing in the consecutive numbering of sub-figures.

Response: We have revised the numbering in Fig.3 of our revised manuscript.

p.13 1.5: Fig 3k (instead of Fig.2k)

Response: We have revised Fig. 2K to Fig. 3j in our revised manuscript.

p.16 1.26: The reference to Fig.4a and S31 makes no sense here. Fig. S4 and S5?

Response: We have revised the Fig. 4a and S31 to Fig. S4 in our revised manuscript.

REVIEWERS' COMMENTS:

Reviewer #1 (Remarks to the Author):

Referee 1. This reviewer (R1) is now completely satisfied the initial raised points, particularly regarding the details for the calculations made. These enhance the manuscript and are good starting point for other authors to reproduce or perform similar calculations.

Therefore, R1 recommends the publication of this manuscript in the present form.

Referee 2. The main concern raised by this reviewer (R2) was... "Elaborate explanation of the effect or contribution of each synthesis step on the stabilization of non-stoichiometric Ni(OH)₂ with 4 coordinated Ni would add to the significance of the study and pave the way as a methodology for systematic formation and control of low coordinated and non-stoichiometric transition metal hydroxides". Followed by punctual and deeply conceptual questions, that in the present version of the answer, the author worked hard showing evidence particularly of the role the NiS₂ precursor exhibits. There is just a mixed notation on the systems H-NiS₂ and M-NiS₂ regarding the conclusion that the oxygen content in the sample influences the performance of the NR-Ni(OH)₂ system. Just make sure this misunderstanding is not passed through the main manuscript.

And a clear remark in the manuscript should be written on, regarding the further studies to be made on the relevance of the oxygen content and the Ni-O bond, relevant for the performance of these MOR catalysts.

The paper can be published after these minor revisions.

Reviewer #3 (Remarks to the Author):

The authors made all the necessary revisions. In my opinion the paper should be published in the present form.

Andreas Glösen

REVIEWER COMMENTS

Reviewer #1 (Remarks to the Author):

Referee 1. This reviewer (R1) is now completely satisfied the initial raised points, particularly regarding the details for the calculations made. These enhance the manuscript and are good starting point for other authors to reproduce or perform similar calculations.

Therefore, R1 recommends the publication of this manuscript in the present form.

Response: N/A

Referee 2. The main concern raised by this reviewer (R2) was... "Elaborate explanation of the effect or contribution of each synthesis step on the stabilization of non-stoichiometric Ni(OH)₂ with 4 coordinated Ni would add to the significance of the study and pave the way as a methodology for systematic formation and control of low coordinated and non-stoichiometric transition metal hydroxides". Followed by punctual and deeply conceptual questions, that in the present version of the answer, the author work hard showing evidence particularly of the role the NiS₂ precursor exhibits. There is just a mixed notation on the systems H-NiS₂ and M-NiS₂ regarding the conclusion that the oxygen content in the sample influences the performance of the NR-Ni(OH)₂ system. Just make sure this misunderstanding is not passed through the main manuscript.

Response: We are thankful for the suggestion. We like to clarify that we did not mention H-NiS₂ and M-NiS₂ in our manuscript. We only used these two terms (H-NiS₂ and M-NiS₂) in the previous response to reviewer's comment letter to explain the difference in the source of NiS₂ precursor. Thus, our manuscript did not contain these two terms, and we have ensured the readability and clarity of our manuscript.

And a clear remark in the manuscript should be written on, regarding the further studies to be made on the relevance of the oxygen content and the Ni-O bond, relevant for the performance of these MOR catalysts.

Response: We appreciate this constructive suggestion. According to the reviewer's suggestion, we have included this remark in the manuscript (the last paragraph of results section, Page 16 in our revised manuscript): "It was reported that there was no formation of NR-Ni(OH)₂ with alternating four-six coordinated Ni atoms when reacting Ni(OH)₂ with S (25). Combining the results obtained in this work, it is realized that the oxygen content and Ni-O bond in NiS₂ may exert certain influence on the formation of NR-Ni(OH)₂, which requires further comprehensive studies".

The paper can be published after these minor revisions.

Response: N/A

Reviewer #3 (Remarks to the Author):

The authors made all the necessary revisions. In my opinion the paper should be published in the present form.

To support the claim that methanol is an attractive fuel due to its high energy density and that “direct-methanol fuel cells (DMFCs), being one such device that utilizes methanol as fuel, has attracted significant research interest due to its high portability and its promise of high energy density” I would like to suggest adding the following references to give a more balanced overview of DMFC research around the world:

1. Gottesfeld, S., Design Concepts and Durability Challenges for Mini Fuel Cells. In Handbook of Fuel Cells, W. Vielstich, A. L., H.A. Gasteiger and H. Yokokawa, Ed. 2010; Vol. 6, pp 762-778.
2. Kang, K.; Park, S.; Cho, S. O.; Choi, K.; Ju, H., Development of Lightweight 200-W Direct Methanol Fuel Cell System for Unmanned Aerial Vehicle Applications and Flight Demonstration. *Fuel Cells* 2014, 14, 694-700.
3. Sgroi, M.; Zedde, F.; Barbera, O.; Stassi, A.; Sebastián, D.; Lufrano, F.; Baglio, V.; Aricò, A.; Bonde, J.; Schuster, M., Cost Analysis of Direct Methanol Fuel Cell Stacks for Mass Production. *Energies* 2016, 9, 1008.

Response: we are thankful for the suggestion. According to the reviewer’s advices, we have replaced references 3-5 using the reviewer’s advices to support the advantages of direct methanol fuel cell. The revised references are shown below:

3. Gottesfeld, S. Design concepts and durability challenges for mini fuel cells. *Handbook of fuel cells – fundamentals, technology and applications* (John Wiley & Sons, Caesarea, 2010).
4. Kang, K., Park, S., Choi, K.& Ju, H. Development of lightweight 200-W direct methanol fuel cell system for unmanned aerial vehicle applications and flight demonstration. *Fuel cells*. **14**, 594-700 (2014).
5. Sgroi, M. et. al. Cost analysis of direct methanol fuel cell stacks for mass production. *Energies* **9**, 1008 (2016).